# Clinical Implication of Sequential Circulating Tumor DNA Assessments for the Treatment of Diffuse Large B-Cell Lymphoma

**DOI:** 10.3390/cancers17111734

**Published:** 2025-05-22

**Authors:** Ga-Young Song, Joo Heon Park, Sae-Ryung Kang, Seung Jung Han, Youjin Jung, Minuk Son, Ho Cheol Jang, Mihee Kim, Seo-Yeon Ahn, Sung-Hoon Jung, Jae-Sook Ahn, Je-Jung Lee, Hyeoung-Joon Kim, Deok-Hwan Yang

**Affiliations:** 1Department of Hematology-Oncology, Chonnam National University Hwasun Hospital, Medical School of Chonnam National University, Hwasun-gun 58128, Jeollanam-do, Republic of Korea; songga0e@naver.com (G.-Y.S.); minjokobject@gmail.com (H.C.J.); mihi4300@gmail.com (M.K.); armful12@hanmail.net (S.-Y.A.); shglory@hanmail.net (S.-H.J.); ahnjaesook@hanmail.net (J.-S.A.); drjejung@chonnam.ac.kr (J.-J.L.); hjoonk@chonnam.ac.kr (H.-J.K.); 2Department of Laboratory Medicine, Chonnam National University Hwasun Hospital, Medical School of Chonnam National University, Hwasun-gun 58128, Jeollanam-do, Republic of Korea; 3Department of Nuclear Medicine, Chonnam National University Hwasun Hospital, Medical School of Chonnam National University, Hwasun-gun 58128, Jeollanam-do, Republic of Korea; campanella9@naver.com; 4Dxome Co. Ltd., Bundang-gu, Seongnam-si 13558, Gyeonggi-do, Republic of Korea; sjhan@dxome.com (S.J.H.); yjjung@dxome.com (Y.J.); muson@dxome.com (M.S.)

**Keywords:** diffuse large B-cell lymphoma, ctDNA

## Abstract

This prospective study evaluated the clinical utility of sequential circulating tumor DNA (ctDNA) monitoring in 52 patients with newly diagnosed advanced-stage diffuse large B-cell lymphoma (DLBCL) treated with R-CHOP. ctDNA was detected in 98.1% of patients at baseline, showing a 74.7% concordance with tumor tissue genotyping. Higher baseline ctDNA levels were associated with poor prognostic features such as elevated LDH, older age, and higher IPI scores. A ≥2-log reduction in ctDNA after three treatment cycles was significantly associated with improved overall survival but not progression-free survival. Integration of interim ctDNA changes with PET/CT response improved prognostic accuracy, particularly in patients with partial metabolic responses. Notably, patients with both ctDNA and PET/CT positivity at interim had significantly worse survival outcomes. The study highlights ctDNA as a feasible non-invasive biomarker for tumor genotyping and dynamic risk stratification in DLBCL. It also emphasizes the need for highly sensitive ctDNA assays, especially for predicting CNS relapse. Overall, combining ctDNA dynamics with imaging could enhance individualized treatment strategies in DLBCL.

## 1. Introduction

Diffuse large B-cell lymphoma (DLBCL) is the most common subtype of non-Hodgkin lymphoma and is considered a highly aggressive malignancy [1]. Despite advancements in treatment, including the introduction of the rituximab, cyclophosphamide, doxorubicin, vincristine, and prednisolone (R-CHOP) immunochemotherapy regimen, the prognosis of DLBCL remains variable, especially in patients with advanced-stage disease. Standard imaging techniques, such as PET/CT, have limitations in accurately predicting outcomes early in treatment, underscoring a need for reliable, non-invasive biomarkers to assess treatment response and improve prognostication.

Circulating tumor DNA (ctDNA) has emerged as a promising tool in oncology for monitoring tumor dynamics in real time. Derived from fragments of tumor DNA shed into the bloodstream, ctDNA offers a non-invasive method to characterize the genetic landscape of tumors [2]. In DLBCL, ctDNA analysis has the potential to detect tumor-specific alterations, track response to therapy, and identify early signs of recurrence [3,4]. Several studies have demonstrated the prognostic value of ctDNA in various cancers, with ctDNA concentrations found to be correlated with tumor burden and treatment response [5,6,7]. However, data addressing the clinical application of ctDNA monitoring specifically for DLBCL are limited.

This study investigated the clinical utility of sequential ctDNA analysis in patients with advanced-stage DLBCL undergoing R-CHOP therapy. By measuring ctDNA concentrations at multiple time points during treatment, this study aimed to establish whether changes in ctDNA can serve as an early indicator of response, offering a more precise and individualized approach to patient management. Additionally, we explored whether the combined use of ctDNA and PET/CT improved the accuracy of DLBCL outcome prediction as a potential predictive biomarker.

## 2. Patients and Methods

### 2.1. Patients and Sample Collection

This study enrolled patients with newly diagnosed DLBCL undergoing treatment at a single institution between 2022 March and 2023 April. All patients had a pathologic diagnosis of DLBCL or primary mediastinal large B-cell lymphoma according to the 2016 WHO criteria, and patients with an antecedent low-grade lymphoma with histologic transformation were considered eligible [8]. Patients with advanced-stage cancers (Stage 3 or 4 according to the Ann Arbor staging for Hodgkin and non-Hodgkin lymphoma) or Stage 2 cancers with bulky disease (≥8 cm) and who were also scheduled for 6 cycles of R-CHOP treatment were included in this study.

Plasma samples were collected at diagnosis before the administration of chemotherapy (baseline), on the first day of the 4th R-CHOP cycle (interim), and at one month after the 6th cycle of R-CHOP (end of treatment). Paired formalin-fixed paraffin-embedded (FFPE) tissue samples were collected at diagnosis from available patients. ^18^F-FDG PET/CT was performed at each plasma sample collection. Treatment response was assessed by the end-of-therapy PET/CT according to the Lugano response criteria for non-Hodgkin lymphoma [9].

This study was approved by the institutional ethics committees of Chonnam National University Hwasun Hospital and conducted in accordance with the Declaration of Helsinki (CNUHH-2022-016). Informed consent was also obtained from all participants.

### 2.2. Next-Generation Sequencing Library

The experimental methods are described in the Appendix A. Briefly, genomic DNA was extracted from plasma, peripheral blood mononuclear cells, and FFPE. Library preparation was performed using the DxSeq Lymphoma & Myeloma ctDNA, which includes 112 lymphoma- and myeloma-related genes (Appendix A) for the Illumina Platform Kit (Dxome, Seongnam-si, Gyeonggi-do, Republic of Korea ) according to the manufacturer’s instructions. Paired-end sequencing was performed on the NovaSeq 6000 System (Illumina, San Diego, CA, USA), and acquired FASTQ data were processed using the PiSeq algorithm and DxSeq software v2.1.0 (Dxome, Seongnam-si, Gyeonggi-do, Republic of Korea) for variant calling and annotation. Variants including copy number variations (CNV) were classified as pathogenic, likely pathogenic, or of unknown significance according to the ACMG/AMP guidelines and/or tiers 1, 2, or 3 according to the AMP/ASCO/CAP guidelines. To distinguish tumor-derived mutations from clonal hematopoiesis of indeterminate potential (CHIP) and germlines, we performed matched germline analysis using paired peripheral blood mononuclear cell (PBMC) samples. For each variant detected in the plasma, we compared the variant allele frequency (VAF) in the PBMCs. Variants with similar or higher VAFs in the PBMCs than in the plasma were considered likely CHIP-related and excluded from downstream analyses. CHIP-associated genes were annotated based on previously published gene lists [10,11]. The ctDNA concentrations were expressed in haploid genome equivalents per L of plasma (hGE/L) and calculated by multiplying the VAF for all mutations used for detection calling by the concentration of ctDNA (pg/mL of plasma) and dividing by 3.3, using the assumption that each haploid genomic equivalent weighs 3.3 pg, as previously described by Scherer et al. More details are described in the Appendix A.

### 2.3. LymphGen Classifier

Samples were classified into MCD, BN2, EZB, and ST2 subtypes using the LymphGen algorithm version 2.0 [12]. The LymphGen tool is an open-access online platform provided by the National Cancer Institute (https://llmpp.nih.gov/lymphgen/index.php, Statistical analysis) (accessed on 18 July 2024).

### 2.4. Statistical Analyses

Correlations between variables were evaluated using Fisher’s exact test. The Mann–Whitney U and Kruskal–Wallis tests were used to compare the ctDNA concentrations or VAF according to the clinical response. Progression-free survival (PFS) was defined as the time period from the date of diagnosis to the date of disease progression or death from any cause. Overall survival (OS) was defined as the time period from the date of diagnosis to the date of death from any cause. Surviving patients without disease progression or death were censored at the date of their last follow-up. The PFS and OS were assessed using the Kaplan–Meier method and compared using the log-rank test. Statistical computations were performed using SPSS^®^ software (ver. 27; SPSS^®^, Chicago, IL, USA) and EZR software (ver.1.68, accessed on 30 June 2024) in the “R” language [13]. A *p*-value < 0.05 was considered statistically significant.

## 3. Results

### 3.1. Patient Characteristics and Responses to Treatment

A prospective series of 52 patients was enrolled in this study between March 2022 and April 2023. The median age of the patients was 70 years (range 28–84), and 59.6% of the patients were male. Four patients (7.5%) were assessed as Stage 2 with bulky disease and 48 patients (92.5%) were stage 3–4. Forty-four patients (84.6%) had elevated lactate dehydrogenase (LDH) and twenty-seven patients (51.9%) were assessed as high risk, according to the International Prognostic Index (IPI). The baseline clinical characteristics of the patients are presented in Table 1.

All except four patients completed six cycles of R-CHOP. Three patients experienced disease progression before completion of the sixth cycle of R-CHOP, and one patient died of pneumonia before the interim response assessment. After three cycles of R-CHOP, the interim response was evaluated by PET/CT in 51 cases, with thirty-seven patients achieving complete response (CR), twelve achieving partial responses (PR), one identified as having stable disease (SD), and one other as having progressive disease (PD). At the end-of-treatment (EOT) response assessment, thirty-eight patients achieved CR, two PR, one SD, and ten had PD. After a median follow-up of 19.9 months (range 5.6–29.1 months), 20 patients experienced progressive disease, of whom 15 had primary refractory disease.

### 3.2. Detection of ctDNA at Diagnosis

At baseline, we sequenced the ctDNA from all patients to identify somatic alterations that could be used for ctDNA quantification and disease monitoring and sequenced the tissue DNA of 48 patients. All but one patient (98.1%) harbored at least one tumor-specific alteration in the plasma ctDNA. Figure 1A,B show the mutational profile of the series, after restricting the data to genes that were mutated in more than 2% of the cases. The most frequently mutated genes were *DNMT3A*, *TET2*, *KMT2C*, *TP53*, and *KMT2D* in the plasma ctDNA and *KMT2D*, *TNFRSF14*, *MYD88*, *FAS*, *B2M*, *CD79B*, *TP53*, and *PRDM1* in the tumor tissue (the complete list of mutations is provided in Appendix A). Several mutations observed exclusively in the plasma, including alterations in TET2, DNMT3A, TP53, and ASXL1, were identified as likely CHIP-related based on their presence in the matched PBMC samples. Although these variants were included in the oncoplot for comprehensive mutation profiling (Figure 1A,B), they were excluded from the ctDNA burden quantification and longitudinal tracking. In addition, CHIP and germline variants were excluded from the concordance analysis presented in Figure 1C. The concordance between tumor tissue DNA and plasma ctDNA mutations was 74.7% and a scatter plot describes the correlation between mutations identified in tumor tissue and plasma (R^2^ = 0.276, adjusted R^2^ = 0.261, *p* < 0.001) (Figure 1C,D). Of the 19 most frequently identified major mutations, 15 (78.9%) were more frequently identified in plasma than in tumor tissue (Figure 1E). Moreover, we were able to classify 56.0% of the cases according to the genetic subtypes of the LymphGen algorithm, as detailed in Appendix A.

The median concentration of baseline ctDNA was 513.0 hGE/L (range 0.2–3425.2 hGE/L). Baseline ctDNA concentration was associated with several clinical prognostic factors (Figure 2): baseline ctDNA concentration was higher in patients with high serum LDH concentrations than in patients with normal LDH concentrations (82.3 hGE/L [range 0.0–3425.2 hGE/L] vs. 2.7 hGE/L [range 0.2–1040.0 hGE/L], respectively; *p* = 0.003). Patients older than 60 years had higher baseline ctDNA concentrations than younger patients (14.9 hGE/L, range 0.4–3095.3 hGE/L vs. 72.1 hGE/L, range 0.0–3425.2 hGE/L, *p* = 0.006). Baseline ctDNA concentrations were also greater in patients with higher IPI scores (IPI 1, 0.4 hGE/L, range 0.2–2.4 hGE/L; IPI 2, 8.5 hGE/L, range 0.2–1040.0 hGE/L; IPI 3, 51.4 hGE/L, range 0.0–1624.0 hGE/L; IPI 4, 115.0 hGE/L, range 2.2–3425.2 hGE/L; IPI 5, 769.7 hGE/L, range 72.1–1973.6 hGE/L, *p* = 0.019). However, baseline ctDNA concentrations did not differ significantly between Stage 2 and Stage 3–4 patients (Stage 2, 263.3 hGE/L, range 0.0–3425.2 hGE/L; Stage 3–4, 2.8 hGE/L, range 0.2–1040.0 hGE/L, *p* = 1.000).

Median baseline meanVAF, maxVAF, and ctDNA concentration did not differ significantly between patients who responded to R-CHOP treatment (achieved CR or PR) and those who did not (had SD or PD; meanVAF, 8.2%, range 0.0–64.7% vs. 14.9%, range 0.5–39.8%, *p* = 0.332; maxVAF, 19.1%, range 0.0–74.2% vs. 29.5%, 0.7–88.2%, *p* = 0.493; ctDNA concentration, 45.9 hGE/L, range 0.2~3425.2 hGE/L vs. 118.2 hGE/L, range 2158.0~2826.0 hGE/L, *p* = 0.199; see Appendix A).

### 3.3. Associations of Interim ctDNA with Treatment Response and Survival

Interim and EOT plasma samples were obtained in 51 and 48 patients, respectively. The median interim plasma ctDNA concentration was 0.1 hGE/L (range 0.0–7.3 hGE/L), and the median EOT ctDNA concentration was 0.5 hGE/L (range 0.0–80.2 hGE/L). Mean and maximum interim VAF values were significantly lower in the responder group patients than the non-responder group patients (interim meanVAF, 0.2, range 0.0~8.0 vs. 0.6, range 0.0~20.1, *p* = 0.017; interim maxVAF, 0.3, range 0.0–15.4 vs. 1.0, range 0.0–39.6, *p* = 0.021) (Figure 3A,B). However, there was no difference between the responder and non-responder groups with respect to EOT VAF (EOT meanVAF 0.2, range 0.0–11.2 vs. 0.6, range 0.0–19.4, *p* = 0.388; EOT maxVAF, 0.3, range 0.0–31.1 vs. 0.9, range 0.0–30.4, *p* = 0.404; see Figure 3D,E). Similar results were obtained in the ctDNA concentration analysis. The median interim ctDNA concentration was lower in the responder group than in the non-responder group (0.1 hGE/L, range 0.0~0.7 hGE/L vs. 0.1 hGE/L, range 0.0~7.3 hGE/L, *p* = 0.021) (Figure 3C). There was no difference between the responder and non-responder groups for median EOT ctDNA concentration (0.4 hGE/L, range 0.0~40.6 hGE/L vs. 3.7 hGE/L, range 0.0~80.1 hGE/L, *p* = 0.198) (Figure 3F).

When we employed a previously reported cutoff—a 2- to 2.5-log reduction in ctDNA during treatment as a molecular response [5], thirty-eight patients (73.1%) achieved a 2-log reduction in ctDNA after three cycles of immunochemotherapy and fourteen (26.9%) did not. Table 2 presents the EOT response according to the achievement of an interim ctDNA reduction > 2 log. Six (16.2%) of the patients who achieved a greater than 2-log reduction in the interim ctDNA concentration at EOT and five patients (33.3%) who did not showed PD at their EOT response assessment (*p* = 0.432).

During the follow-up period, 20 patients experienced the disease progression; of these, 18 relapsed within 12 months after completion of six cycles of R-CHOP. The PFS of patients with ≥2-log reduction in interim ctDNA concentration did not differ from that of the patients with <2-log reduction in interim ctDNA concentration (23.6 months vs. not reached, *p* = 0.286). However, the patients with ≥2-log reduction in their interim ctDNA concentration showed a better OS than the patients who did not (not reached vs. not reached, *p* = 0.004; see Appendix A).

Among the mutations detected at diagnosis, the dynamic pattern of *MYD88 L252P* revealed a close correlation with the patient’s response to treatment. In twelve patients who had a *MYD88 L252P* mutation at diagnosis, all except one patient showed a concordant result of the EOT treatment response and *MYD88 L252P* identification at interim or EOT sample (Appendix A).

### 3.4. Combined Interim Response Assessment Incorporating ctDNA and PET/CT

In the interim response assessments combining PET/CT imaging (based on the Lugano classification) and a ctDNA cut-off of a 2-log-fold change, 35 patients had concordant PET/CT and ctDNA responses and 16 had discordant results. Four of twenty-nine patients (13.8%) who achieved both metabolic and molecular responses (PET negative and ctDNA negative) at the interim assessment had relapsed at the EOT assessment, while four of six patients (66.7%) who did not achieve either a metabolic or molecular response (PET positive and ctDNA positive) relapsed (*p* = 0.064). Representative cases are described in Figure 4. In detail, among the four patients in the PET-negative and ctDNA-negative group who relapsed, two had CNS relapse (one isolated CNS relapse and one systemic and CNS relapse simultaneously). Moreover, none of the eight patients who achieved an interim metabolic response but a <2-log reduction in ctDNA (PET negative and ctDNA positive) experienced disease progression. Two of the eight patients who did not achieve an interim metabolic response but showed a ≥2-log reduction in ctDNA (PET positive and ctDNA negative) showed progressive disease, and both had isolated CNS relapse.

The PFS and OS of the six patients with an interim PET-positive and ctDNA-positive response were significantly worse than those of the other patients (median PFS 8.4 months vs. 22.3 months, *p* < 0.001; median OS 13.4 months vs. 27.5 months, *p* < 0.001) (Appendix A). In a subgroup analysis of 13 patients with interim PET partial responses, the PFS and OS differed significantly between those who achieved a 2-log reduction in interim ctDNA concentration at the EOT assessment and those who did not (median PFS not reached vs. 4.0 months, *p* = 0.120; median OS not reached vs. not reached, *p* = 0.044; see Appendix A).

## 4. Discussion

In this study, we evaluated the utility of targeted deep sequencing for the genotyping of ctDNA in newly diagnosed DLBCL patients and assessed their response to R-CHOP chemotherapy. We were able to identify 225 mutations that enable treatment response monitoring in diagnostic plasma samples. We were able to identify ctDNA in 98.1% of our study patients with a high tumor burden, and most of the genetic abnormalities identified in the ctDNA samples were concordant with mutations identified in tumor tissue genotyping. The 74.7% concordance between pretreatment ctDNA and tumor genotyping in this study is comparable to previously reported results from large lymphoma studies [14,15,16,17,18]. Based on the pretreatment ctDNA and tumor genotyping results in our study, genetic subtypes were identified in 75% of the patients using LymphGen classifications. The proportions of the various genetic subtypes were similar to those found in previous reports, although the proportions of EZB and MCD subtypes in our cohort were lower [12]. After excluding four patients who lacked diagnostic tumor samples, forty-six out of forty-eight (95.8%) patients had concordant results with respect to the mutations identified in the tumor and ctDNA, while additional mutations were identified in the ctDNA samples but not in the tumor samples in twenty-nine out of forty-eight (60.4%) patients. Furthermore, in four patients without an available tumor sample, more than one driver mutation was identified in the pretreatment ctDNA sample. These results suggest that non-invasive pretreatment plasma ctDNA can provide detailed tumor genotyping in terms of spatial heterogeneity, and may complement tumor tissue genotyping and possibly replace it in the future.

The prognostic significance of pretreatment ctDNA concentrations has been reported in many previous studies. Pretreatment ctDNA concentration correlates with tumor volume, which is measured as metabolic tumor volume on PET/CT images, and higher pretreatment ctDNA concentrations are associated with worse responses to treatment [5,14,19,20]. In this study, pretreatment ctDNA concentration did not differ between the treatment responder and non-responder groups. One explanation for this result is that all the patients included in this study had advanced disease.

This interpretation is supported by our finding that the median baseline ctDNA concentration was greater than the concentrations reported in other studies [5,14]. In terms of ctDNA dynamics during treatment, the difference in PFS between the patients with or without a ≥2-log reduction in interim ctDNA was not statistically significant. In addition, the median EOT ctDNA concentration did not correlate with the response to immunochemotherapy in this study. The small sample size and low depth of ctDNA detection may have contributed to this result. According to a large retrospective analysis of ctDNA results for 230 DLBCL patients, improving a limit of detection for ctDNA positivity down to 10^−6^ could show superior predictive power for PFS after the second cycle of chemotherapy [21]. Considering recently published reports about the strong predictability of the dynamic ctDNA level as an MRD monitoring tool, more sensitive assays would be needed to monitor MRD detection and predict outcome.

In the present study, the majority of the patients with a final progressive disease had isolated CNS relapse and the result partially explained the lack of correlation between the interim ctDNA level and the final response in this study. The detection of plasma ctDNA is less consistent in CNS lymphoma than in systemic lymphoma. In a study including 92 isolated CNS lymphoma patients, ctDNA was identified in only 78% of plasma samples even when using ultrasensitive panel-directed sequencing [22], suggesting that it might be difficult to predict CNS relapse using less sensitive plasma ctDNA monitoring. Recently, several research works including a small number of DLBCL patients with CNS involvement reported a higher detection yield of ctDNA in cerebrospinal fluid (CSF) than in plasma [23,24]. CSF ctDNA identification is reported to be at the same time or earlier than the clinical CNS relapse, and the paired analysis of plasma and CSF ctDNA might improve the prediction of CNS relapse [25]. Patients who are at high risk of CNS relapse received a CNS prophylaxis of either intrathecal or intravenous methotrexate during the six cycles of R-CHOP in this study. Among the seven patients with CNS relapse, four patients did not receive prophylactic methotrexate because they were assessed at low or intermediate risk of CNS relapse. The high CNS relapse rate in the present study despite the CNS prophylaxis suggests an unmet need of more sensitive methods to predict the CNS relapse, and CSF ctDNA analysis could improve the current CNS prophylaxis strategy in DLBCL.

Although the present study results did not show statistical significance between the ctDNA reduction rate and the treatment response in all patients, in a subgroup analysis of patients who had MYD88 L252P at diagnosis, the ctDNA response was correlated with treatment response. MYD88 mutations are often observed in paired diagnostic and relapsed samples, suggesting they occur early in the disease process and persist through relapses in certain types of lymphoma [26]. Previous research demonstrated that MYD88 identification in liquid biopsies could aid in monitoring the disease course in patients with DLBCL and Waldenstrom’s macroglobulinemia [27,28]. In addition to these previous results, the finding about MYD88 in this study suggests the possibility of the early identification of patients with poor prognosis through ctDNA tracking.

In subgroup analyses of the patients who achieved PR at the interim PET/CT assessment, the PFS and OS differed significantly depending on the interim ctDNA reduction ≥ 2-log-fold. This is meaningful in that the ctDNA responses further stratify the prognosis of the patients with PR in the interim PET/CT response whose prognoses are heterogeneous. A combined ctDNA and PET/CT response assessment during treatment might improve the predictive ability of treatment responses and serve as a background for studies about treatment modification according to interim response.

This study highlights the clinical utility of ctDNA as a non-invasive biomarker for genotyping and monitoring treatment response in patients with advanced-stage DLBCL. Our ctDNA analysis demonstrated a high concordance with tumor tissue genotyping and effectively captured tumor heterogeneity. While pretreatment ctDNA concentrations were correlated with several clinical prognostic factors, a ≥ 2-log reduction in interim ctDNA concentrations strongly predicted improved OS. Integrating ctDNA analysis with PET/CT imaging further enhanced the prediction of treatment outcomes, particularly by stratifying the prognosis in patients with partial metabolic responses on PET/CT. However, our study found only a modest predictive value for PFS and identified challenges in detecting CNS relapse that underscore the need for more sensitive analytical techniques and longer follow-up periods. Overall, the study findings could suggest ctDNA as a valuable complement to standard imaging modalities, enabling personalized treatment approaches and the improved management of DLBCL.

## 5. Conclusions

This study demonstrates that sequential ctDNA monitoring is a clinically valuable tool for non-invasive tumor genotyping and dynamic response assessment in advanced-stage DLBCL. Integrating ctDNA dynamics with PET/CT enhances risk stratification, especially in patients with partial metabolic responses. These findings support the incorporation of ctDNA analysis into routine clinical practice to guide personalized therapeutic strategies in DLBCL.

## Figures and Tables

**Figure 1 cancers-17-01734-f001:**
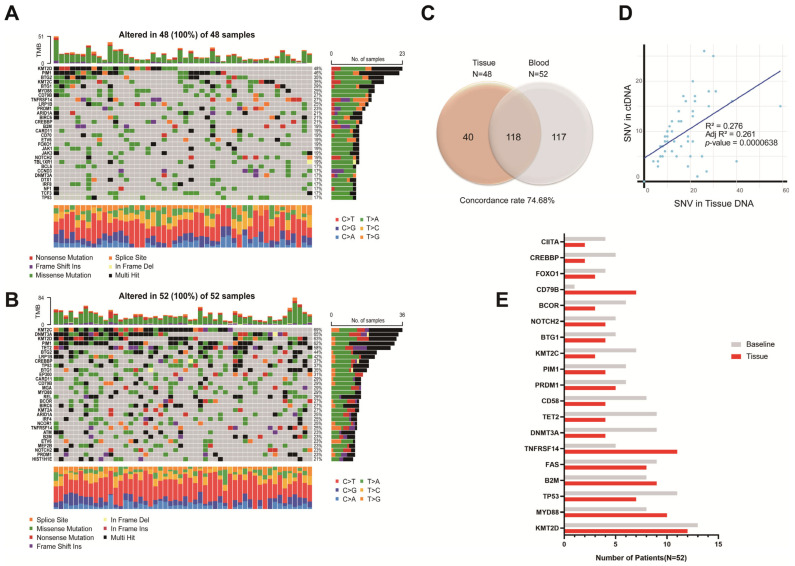
Mutational profile of DNA in tumor (**A**) and plasma (circulating-tumor DNA [ctDNA]; (**B**) samples of the 52 patients with diffuse large B-cell lymphoma (DLBCL). Each column represents one tumor/plasma sample, and each row represents one gene. Mutations in the heatmap described all the identified mutations including tier 1 to tier 3. Mutations identified in tumor and plasma samples from individual DLBCL patients showed 74.68% concordance (**C**); the correlation between mutations identified in tumor tissue and plasma is shown in (**D**); the frequency of 19 major mutations in plasma and tumor samples are presented in (**E**).

**Figure 2 cancers-17-01734-f002:**
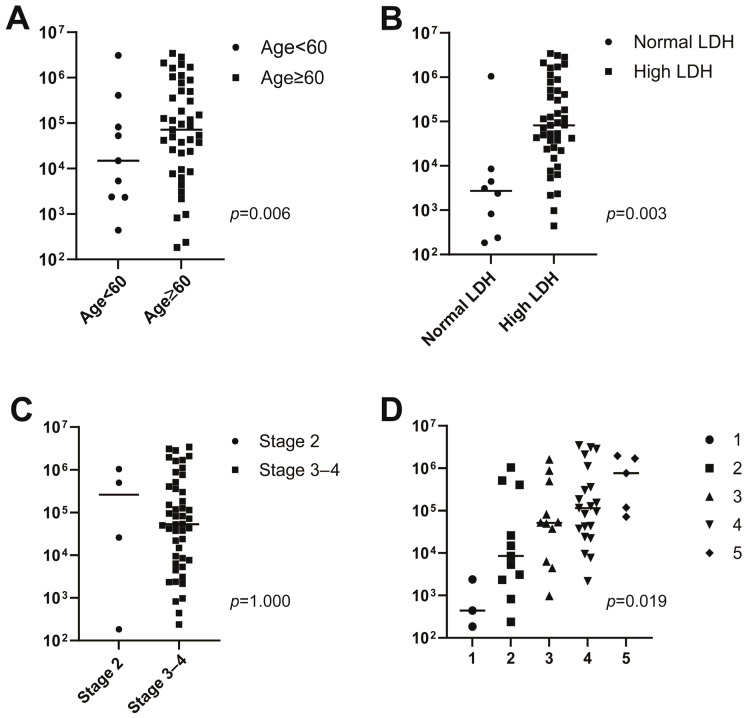
Associations between baseline circulating tumor DNA (ctDNA) concentration and age (**A**), lactate dehydrogenase (LDH) (**B**), stage (**C**), and the International Prognostic Index (IPI) (**D**).

**Figure 3 cancers-17-01734-f003:**
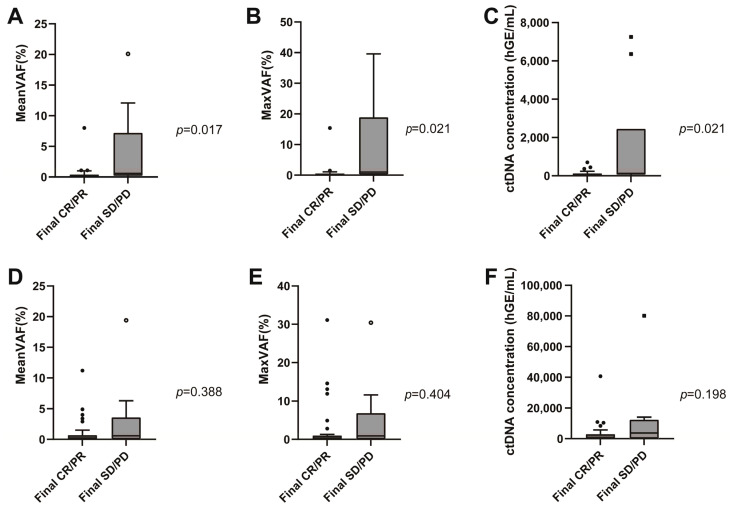
Interim meanVAF (**A**), maxVAF (**B**), median circulating tumor DNA (ctDNA) concentration (**C**) according to the patient response to frontline immunochemotherapy. End-of-treatment meanVAF (**D**), maxVAF (**E**), and median ctDNA concentration (**F**) according to the response to frontline immunochemotherapy.

**Figure 4 cancers-17-01734-f004:**
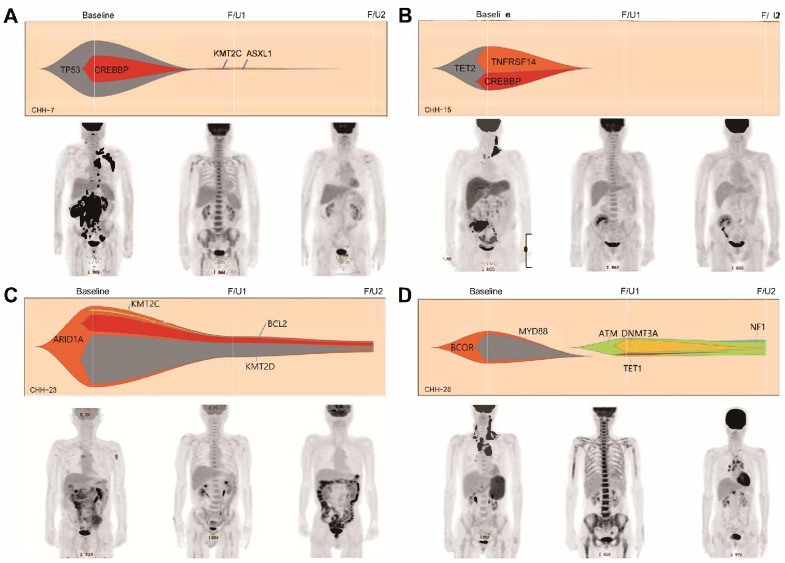
Representative cases emphasizing the clinical usefulness of ctDNA as a response assessment tool complementary to positron emission tomography/computed tomography (PET/CT). The first two cases show concordant results between the interim PET/CT response and the ctDNA response assessment. Those patients are being followed up without relapse (**A**,**B**). In case C, the patients achieved an interim complete metabolic response in PET/CT but the interim ctDNA analysis result showed the persistence of driver mutations which was identified at pretreatment. The patient experienced disease relapse at the final response assessment (**C**). The last case also presents the interim complete metabolic response with bone marrow reactivation. However, new mutations which were not identified in the pretreatment sample were identified and persistently identified in the final sample, and the final PET/CT revealed disease relapse (**D**). In Figure 1D, F/U1 and F/U2 were applied at a magnified scale (20×).

**Table 1 cancers-17-01734-t001:** Baseline clinical characteristics of the 52 patients with diffuse large B-cell lymphoma enrolled in the study.

	N = 52
Age, median (range)	70 (28–84)
Sex, *n* (%)	
Male	31 (59.6%)
Female	21 (40.4%)
High LDH, *n* (%)	44 (84.6%)
Stage, *n* (%)	
2 bulky	4 (7.6%)
3–4	48 (92.4%)
PS ≥ 2, *n* (%)	15 (28.8%)
B-Symptom, *n* (%)	20 (38.5%)
IPI, *n* (%)	
1–2	13 (25.0%)
3	12 (23.1%)
4–5	27 (51.9%)

**Table 2 cancers-17-01734-t002:** Final response assessment according to interim ctDNA concentration (cutoff > 2-log reduction) among the 52 patients with diffuse large B-cell lymphoma enrolled in the study.

Final Response	Interim ctDNA ≥ 2-log Reduction (*n* = 37)	Interim ctDNA < 2-log Reduction (*n* = 15)	*p*-Value
CR (*n* = 37)	28 (75.7%)	9 (60.0%)	0.724
PR (*n* = 2)	2 (5.4%)	0 (0.0%)	
SD (*n* = 1)	1 (2.7%)	0 (0.0%)	
PD (*n* = 11)	6 (16.2%)	5 (33.3%)	
NA (*n* = 1)	0 (0.0%)	1 (6.6%)	

## Data Availability

The datasets generated during and/or analyzed during the current study are available from the corresponding author upon reasonable request.

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
