# Peer review of "Clinical Implication of Sequential Circulating Tumor DNA Assessments for the Treatment of Diffuse Large B-Cell Lymphoma"

_cancers, 2025, doi:10.3390/cancers17111734_

Round 1

Reviewer 1 Report

Comments and Suggestions for Authors

In this manuscript, Dr. Song and coauthors aim to study the utility of circulating tumor DNA monitoring for prognosis and response assessment in diffuse large B-cell lymphoma. The paper is well written and presented. While there are few surprises the authors are able to assess the value and limitations of ctDNA monitoring in this use case. I do have a few small comments that the authors need to address.

1) Some of the mutations detected exclusively in plasma are almost certainly clonal hematopoiesis of indeterminant potential (CHIP). TET2 in particular but others as well definitely fall in the category. This should be addressed. The expected rate of CHIP clone detection should be fairly high given the age of this population. Regardless the authors should make sure this signal is removed from the DLBCL ctDNA signal as it is an unrelated phenomenon.

2) CHIP clones often have prognostic significance for patients. An advantage of ctDNA diagnostics is the identification of these clones, even if they are not a part of the tumor signal. If the authors can show that there is a prognostic impact, that would be one more reason to consider their approach.

3) MYD88 mutations in non-Hodgkin’s lymphomas including CLL, Waldenstroms Macroglobulinemia and DLBCL, tend to be very early if not founding clonal events when they occur. This means VAF tracking can be used as a proxy for tumor burden. This has been studied by others and warrants at least a mention in the discussion.

Author Response

Reviewer 1

In this manuscript, Dr. Song and coauthors aim to study the utility of circulating tumor DNA monitoring for prognosis and response assessment in diffuse large B-cell lymphoma. The paper is well written and presented. While there are few surprises the authors are able to assess the value and limitations of ctDNA monitoring in this use case. I do have a few small comments that the authors need to address.

1) Some of the mutations detected exclusively in plasma are almost certainly clonal hematopoiesis of indeterminant potential (CHIP). TET2 in particular but others as well definitely fall in the category. This should be addressed. The expected rate of CHIP clone detection should be fairly high given the age of this population. Regardless the authors should make sure this signal is removed from the DLBCL ctDNA signal as it is an unrelated phenomenon.

[Answer]

We appreciate the reviewer’s insightful comment. As suggested, we carefully reviewed the plasma-only mutations and identified a subset—including TET2, DNMT3A, TP53 and ASXL1—that are commonly associated with clonal hematopoiesis of indeterminate potential (CHIP). While these mutations were included in the initial oncoplot for comprehensive visualization, they were excluded from downstream ctDNA burden estimation and longitudinal analyses.

To distinguish CHIP & germline mutation from tumor-derived variants, we performed matched germline analysis using paired PBMC samples. Variants with comparable or higher variant allele frequencies (VAFs) in the PBMCs relative to plasma were classified as likely CHIP-related and excluded from tumor-specific ctDNA tracking. This filtering strategy is now described in the Methods section, and relevant details have been added to the Results section.

Methods section (Page 4, Line 16-22)

To distinguish tumor-derived mutations from clonal hematopoiesis of indeterminate potential (CHIP) and germline, we performed matched germline analysis using paired peripheral blood mononuclear cell (PBMC) samples. For each variant detected in plasma, we compared the variant allele frequency (VAF) in PBMCs. Variants with similar or higher VAFs in PBMCs than in plasma were considered likely CHIP-related and excluded from downstream analyses. CHIP-associated genes were annotated based on previously published gene lists (e.g., Genovese et al., NEJM 2014; Jaiswal et al., NEJM 2014).

Results section (Page 6, Line 25-30)

Several mutations observed exclusively in plasma, including alterations in TET2, DNMT3A, TP53 and ASXL1, were identified as likely CHIP-related based on their presence in matched PBMC samples. Although these variants were included in the oncoplot for comprehensive mutation profiling (Figure 1A and 1B), they were excluded from ctDNA burden quantification and longitudinal tracking. In addition, CHIP and germline variants were excluded from the concordance analysis presented in Figure 1C.

2) CHIP clones often have prognostic significance for patients. An advantage of ctDNA diagnostics is the identification of these clones, even if they are not a part of the tumor signal. If the authors can show that there is a prognostic impact, that would be one more reason to consider their approach.

[Answer]

The authors agree with the reviewer’s opinion. CHIP clones were identified at diagnosis according to ctDNA analysis in 10/52 (19.2%) of the patients included in the present study (DNMT3A 11.5%, TET2 9.6%, ASXL3 1.9%). Some studies have revealed that DLBCL patients harboring CHIP had inferior survival outcome (Liu et al. Haematologica. 2023, Amini et al. Am J Hematol 2020). However, we could not find any correlation between baseline clinical prognostic factors (age, LDH, stage, IPI), concentration of ctDNA, survival outcomes and CHIP mutation status. Therefore, as mentioned in the previous answer, CHIP-related mutations were excluded from analysis of baseline ctDNA concentration and response assessment during treatment.

3) MYD88 mutations in non-Hodgkin’s lymphomas including CLL, Waldenstroms Macroglobulinemia and DLBCL, tend to be very early if not founding clonal events when they occur. This means VAF tracking can be used as a proxy for tumor burden. This has been studied by others and warrants at least a mention in the discussion.

[Answer]

The authors sincerely appreciate the thoughtful insights provided by the reviewer. Descriptions related to MYD88 have been added to the Discussion section.

Page 10, Line 6-15

Although the present study results did not show statistical significance between ctDNA reduction rate and treatment response in all patients, in a subgroup analysis of patients who had MYD88 L252P at diagnosis, ctDNA response was correlated with treatment response. MYD88 mutations are often observed in paired diagnostic and relapsed samples, suggesting they occur early in the disease process and persist through relapses in certain types of lymphoma [21]. Previous researches demonstrated that MYD88 identification in liquid biopsies could aid in monitoring disease course in patients with DLBCL and waldenstrom’s macroglobulinemia [22,23]. In addition to these previous results, the finding about MYD88 in this study suggests the possibility of early identification of patients with poor prognosis through ctDNA tracking.

Reviewer 2 Report

Comments and Suggestions for Authors

Review of cancers-3626921 circulating tumor DNA DLBCL

A well-done study with important findings on using PET imaging combined with circulating tumor DNA to prognosticate in DLBCL

I have two major questions - suggestions

  • Why is a 2 log reduction in ctDNA the “gold standard”? I suggest you look at different levels of reduction to with receiver – operator curves to determine the the ctDNA reduction that gives the highest true positive and lowest false positive results for your data.  Then rerun your analysis with the new ctDNA cut off.
  • I think you should expand your discussion of CSF tumor DNA and to include:
    1. Should every patient have a lumber puncture with CSF cytology and tumor DNA at diagnosis?
      1. Kim, S., Kim, J., Park, M., Park, B., Ryu, K., Yoon, S., Kim, W., Shin, S., Lee, S., & , (2024). Feasibility of Circulating Tumor DNA Detection in the Cerebrospinal Fluid of Patients With Central Nervous System Involvement in Large B-Cell Lymphoma. Annals of Laboratory Medicine,
      2. Liang, J., Wu, Y., Shen, H., Li, Y., Liang, J., Gao, R., Hua, W., Shang, C., Du, K., Xing, T., Zhang, X., Wang, C., Zhu, L., Shao, Y., Li, J., Wu, J., Yin, H., Wang, L., & Xu, W. (2024). Clinical implications of CSF-ctDNA positivity in newly diagnosed diffuse large B cell lymphoma. Leukemia, 38(7), 1541-1552.
    2. Should all patients have intrathecal chemotherapy like childhood ALL to prevent CNS seeding from lumbar Puncture?
      1. Fu, H., Wang, T., Yang, Y., Qiu, C., Wang, H., Qiu, Y., Liu, J., & Liu, T. (2025). Next-generation sequencing of circulating tumor DNA in cerebrospinal fluid for detecting gene mutations in central nervous system lymphoma patients. Therapeutic Advances in Hematology, 16, 1
      2. González-Barca, E., group, G.T.O.B.O., Canales, M., Salar, A., Ferreiro-Martínez, J., Ferrer-Bordes, S., García-Marco, J., Sánchez-Blanco, J., García-Frade, J., Peñalver, J., Bello-López, J., Sancho, J., Caballero, D., & , (2016). Central nervous system prophylaxis with intrathecal liposomal cytarabine in a subset of high-risk patients with diffuse large B-cell lymphoma receiving first line systemic therapy in a prospective trial. Annals of Hematology,
      3. The text is difficult to read due to the huge string of numbers with up to 7 significant numbers. Can you change it to produce a minimum of 3 or 4 significant numbers?  

            For example, if you report per Liter rather than per mL the sentence below:

The median concentration of baseline ctDNA was 513,192.9 hGE/mL (range 183.5– 3,425,240.9 hGE/mL).

Becomes this much less cluttered sentence:

The median concentration of baseline ctDNA was 513 hGE/L (range 0.183– 3,425 hGE/L).

  1. Regarding Figure 2, the caption is minimal and I don’t understand what the x-axis of 2D represents. What is 1,2,3,4,5?

Author Response

Reviewer 2

A well-done study with important findings on using PET imaging combined with circulating tumor DNA to prognosticate in DLBCL

I have two major questions - suggestions

  • Why is a 2 log reduction in ctDNA the “gold standard”? I suggest you look at different levels of reduction to with receiver – operator curves to determine the the ctDNA reduction that gives the highest true positive and lowest false positive results for your data.  Then rerun your analysis with the new ctDNA cut off.

[Answer]

Thank you for the valuable statistical insights. We performed ROC analysis to determine the cut-off value of interim/baseline ctDNA reduction. The determined cut-off value was 2.755log with sensitivity of 0.548 and specificity of 0.715. However, this cut-off could not significantly differentiate the survival outcomes of the patients as shown below (PFS, p=0.263; OS, p=0.862). Therefore, the authors adopted previously reported cut-off point of 2log reduction. We absolutely agree with the reviewer’s comment that 2-log reduction could not be assumed to be the gold standard, and further researches might be needed to determine the best cut-off value of ctDNA reduction with large number of patients and more sensitive sequencing method.

  • I think you should expand your discussion of CSF tumor DNA and to include:
  1. Should every patient have a lumber puncture with CSF cytology and tumor DNA at diagnosis?
  2. Kim, S., Kim, J., Park, M., Park, B., Ryu, K., Yoon, S., Kim, W., Shin, S., Lee, S., & , (2024). Feasibility of Circulating Tumor DNA Detection in the Cerebrospinal Fluid of Patients With Central Nervous System Involvement in Large B-Cell Lymphoma. Annals of Laboratory Medicine,
  3. Liang, J., Wu, Y., Shen, H., Li, Y., Liang, J., Gao, R., Hua, W., Shang, C., Du, K., Xing, T., Zhang, X., Wang, C., Zhu, L., Shao, Y., Li, J., Wu, J., Yin, H., Wang, L., & Xu, W. (2024). Clinical implications of CSF-ctDNA positivity in newly diagnosed diffuse large B cell lymphoma. Leukemia, 38(7), 1541-1552.
  4. Should all patients have intrathecal chemotherapy like childhood ALL to prevent CNS seeding from lumbar Puncture?
  5. Fu, H., Wang, T., Yang, Y., Qiu, C., Wang, H., Qiu, Y., Liu, J., & Liu, T. (2025). Next-generation sequencing of circulating tumor DNA in cerebrospinal fluid for detecting gene mutations in central nervous system lymphoma patients. Therapeutic Advances in Hematology, 16, 1
  6. González-Barca, E., group, G.T.O.B.O., Canales, M., Salar, A., Ferreiro-Martínez, J., Ferrer-Bordes, S., García-Marco, J., Sánchez-Blanco, J., García-Frade, J., Peñalver, J., Bello-López, J., Sancho, J., Caballero, D., & , (2016). Central nervous system prophylaxis with intrathecal liposomal cytarabine in a subset of high-risk patients with diffuse large B-cell lymphoma receiving first line systemic therapy in a prospective trial. Annals of Hematology,

[Answer]

Thank you for the constructive feedback. As reviewer mentioned, the authors revised the discussion section as follows;

Page 9, Line 36-Page 10, Line 5

Recently, several researches including small number of DLBCL patients with CNS involvement reported higher detection yield of ctDNA in cerebrospinal fluid (CSF) than in plasma[23,24]. CSF ctDNA identification is reported to be at the same time or earlier than clinical CNS relapse and paired analysis of plasma and CSF ctDNA might improve the prediction of CNS relapse[25]. Patients who are at high risk of CNS relapse received CNS prophylaxis of either intrathecal or intravenous methotrexate during 6 cycles of R-CHOP in this study. Among the 7 patients with CNS relapse, 4 patients did not recieved prophylactic methotrexate because they were assessed low or intermediate risk of CNS relapse. High CNS relapse rate in the present study despite of CNS prophylaxis suggests unmet need of more sensitive methods to predict the CNS relapse, and CSF ctDNA analysis could improve the current CNS prophylaxis strategy in DLBCL.

  • The text is difficult to read due to the huge string of numbers with up to 7 significant numbers. Can you change it to produce a minimum of 3 or 4 significant numbers?  For example, if you report per Liter rather than per mL the sentence below:The median concentration of baseline ctDNA was 513,192.9 hGE/mL (range 183.5– 3,425,240.9 hGE/mL). Becomes this much less cluttered sentence:The median concentration of baseline ctDNA was 513 hGE/L (range 0.183– 3,425 hGE/L).

[Answer] The authors acknowledge the reviewer’s comment, and we have revised the method and result section as follows;

Method section (Page 4, Line 22-26)

The ctDNA concentrations were expressed in haploid genome equivalents per L of plasma (hGE/L) and calculated by multiplying the VAF for all mutations used for detection calling by the concentration of ctDNA (pg/mL of plasma) and dividing by 3.3, using the assumption that each haploid genomic equivalent weighs 3.3 pg, as previously described by Scherer et al. More details are described in Supplementary material.

Result section (Page 6, Line 37-Page 7, Line 30)

The median concentration of baseline ctDNA was 513.0 hGE/L (range 0.2–3,425.2 hGE/L). Baseline ctDNA concentration was associated with several clinical prognostic factors (Figure 2): baseline ctDNA concentration was higher in patients with high serum LDH concentrations than in patients with normal LDH concentrations (82.3hGE/L [range 0.0–3,425.2hGE/L] vs. 2.7hGE/L [range 0.2–1,040.0hGE/L], respectively; p = 0.003). Patients older than 60 years had higher baseline ctDNA concentrations than younger patients (14.9hGE/L, range 0.4-3,095.3hGE/L vs. 72.1hGE/L, range 0.0-3,425.2hGE/L, p=0.006). Baseline ctDNA concentrations were also greater in patients with higher IPI scores (IPI 1, 0.4hGE/L, range 0.2-2.4hGE/L; IPI 2, 8.5hGE/L, range 0.2-1,040.0hGE/L; IPI 3, 51.4hGE/L, range 0.0-1,624.0 hGE/L; IPI 4, 115.0hGE/L, range 2.2-3,425.2hGE/L; IPI 5, 769.7hGE/L, range 72.1-1,973.6hGE/L, p=0.019). However, baseline ctDNA concentrations did not differ significantly between Stage 2 and Stage 3–4 patients (Stage 2, 263.3hGE/L, range 0.0-3,425.2 hGE/L; Stage 3-4, 2.8hGE/L, range 0.2-1,040.0hGE/L, p=1.000).

Median baseline meanVAF, maxVAF, and ctDNA concentration did not differ significantly between patients who responded to R-CHOP treatment (achieved CR or PR) and those who did not (had SD or PD; meanVAF, 8.2%, range 0.0-64.7% vs. 14.9%, range 0.5-39.8%, p=0.332; maxVAF, 19.1%, range 0.0-74.2% vs. 29.5%, 0.7-88.2%, p=0.493; ctDNA concentration, 45.9hGE/L, range 0.2~3,425.2 hGE/L vs. 118.2 hGE/L, range 2,158.0~2,826.0hGE/L, p=0.199; see Figure S2A-C).

Associations of interim ctDNA with treatment response and survival

Interim and EOT plasma samples were obtained in 51 and 48 patients, respectively. The median interim plasma ctDNA concentration was 0.1 hGE/L (range 0.0-7.3 hGE/L), and the median EOT ctDNA concentration was 0.5 hGE/L (range 0.0-80.2 hGE/L). Mean and maximum interim VAF was significantly lower in responder group patients than non-responder group patients (interim meanVAF, 0.2, range 0.0~8.0 vs. 0.6, range 0.0~20.1, p=0.017; interim maxVAF, 0.3, range 0.0-15.4 vs. 1.0, range 0.0-39.6, p=0.021) (Figure 3A, B). However, there was no difference between the responder and non-responder groups with respect to EOT VAF (EOT meanVAF 0.2, range 0.0-11.2 vs. 0.6, range 0.0-19.4, p=0.388; EOT maxVAF, 0.3, range 0.0-31.1 vs. 0.9, range 0.0-30.4, p=0.404; see Figure 3D, E). Similar results were obtained in the ctDNA concentration analysis. Median interim ctDNA concentration was lower in the responder group than in the non-responder group (0.1hGE/L, range 0.0~0.7 hGE/L vs. 0.1 hGE/L, range 0.0~7.3 hGE/L, p=0.021) (Figure 3C). There was no difference between responder and non-responder groups for median EOT ctDNA concentration (0.4 hGE/L, range 0.0~40.6 hGE/L vs. 3.7 hGE/L, range 0.0~80.1 hGE/L, p=0.198) (Figure 3F).

  • Regarding Figure 2, the caption is minimal and I don’t understand what the x-axis of 2D represents. What is 1,2,3,4,5?

[Answer]

The authors revised the caption of Figure 2 as follows;

Figure legend (Page 14, Line 1-2)

Figure 2. Associations between baseline circulating tumor DNA (ctDNA) concentration and age (A), lactate dehydrogenase (LDH) (B), stage (C), and the International Prognostic Index (IPI) (D).

Round 2

Reviewer 1 Report

Comments and Suggestions for Authors

Thank you for the thoughtful response. While I personally would have mentioned the CHIP detection more in the discussion as a positive aspect of liquid biopsy as they can be of clinical interest even if they did not correlate with prognosis in this study. That bing said, it is fine as is. My only other suggestion is to consider switching to log scale for figure 3 since a few outliers make it hard to see the very low values where most of the action is.

Reviewer 2 Report

Comments and Suggestions for Authors

Thank you for your response to my comments.